# Examining Anxiety, Sleep Quality, and Physical Activity as Predictors of Depression among University Students from Saudi Arabia during the Second Wave of the COVID-19 Pandemic

**DOI:** 10.3390/ijerph19106262

**Published:** 2022-05-21

**Authors:** Tahani K. Alshammari, Aljawharah M. Alkhodair, Hanan A. Alhebshi, Aleksandra M. Rogowska, Awatif B. Albaker, Nouf T. AL-Damri, Anfal F. Bin Dayel, Asma S. Alonazi, Nouf M. Alrasheed, Musaad A. Alshammari

**Affiliations:** 1Department of Pharmacology and Toxicology, College of Pharmacy, King Saud University, Riyadh 11451, Saudi Arabia; 437200640@student.ksu.edu.sa (A.M.A.); 437202534@student.ksu.edu.sa (H.A.A.); abaker@ksu.edu.sa (A.B.A.); naldamri@ksu.edu.sa (N.T.A.-D.); abindayel@ksu.edu.sa (A.F.B.D.); aaloneazi@ksu.edu.sa (A.S.A.); nrasheed@ksu.edu.sa (N.M.A.); malshammari@ksu.edu.sa (M.A.A.); 2Institute of Psychology, University of Opole, 45-040 Opole, Poland; arogowska@uni.opole.pl

**Keywords:** depression, anxiety, COVID-19, sleep quality, mental health, university counseling services, Saudi Arabia

## Abstract

Conducted during the second wave of the pandemic, this cross-sectional study examined the link between sleep quality, physical activity, exposure, and the impact of COVID-19 as predictors of mental health in Saudi undergraduate students. A convenience sample of 207 participants were recruited, 89% of whom were females and 94% were single. The measures included questionnaires on the level of exposure and the perceived impact of COVID-19, a physical activity measure, GAD-7, PHQ-9, and PSQI. The results indicated that approximately 43% of participants exhibited moderate anxiety, and 50% were at risk of depression. Overall, 63.93% of students exposed to strict quarantine for at least 14 days (*n* = 39) exhibited a high risk of developing depression (χ^2^(1) = 6.49, *p* < 0.05, ϕ = 0.18). A higher risk of depression was also found in students whose loved ones lost their jobs (χ^2^(1) = 4.24, *p* < 0.05, ϕ = 0.14). Moreover, there was also a strong association between depression and anxiety (β = 0.33, *p* < 0.01), sleep quality (β = 0.32, *p* < 0.01), and the perceived negative impact of COVID-19 on socio-economic status (β = 0.26, *p* < 0.05), explaining 66.67% of depression variance. Our study highlights the socio-economic impact of this pandemic and the overwhelming prevalence of depression.

## 1. Introduction

According to the World Health Organization (WHO), the 2019 (COVID-19) outbreak posed a severe public health threat worldwide [1]. In response to this pandemic, most governments implemented instructions and policies to prevent the spread of the disease by imposing lockdowns, social distancing, home quarantine, and the sudden transition of the entire education system to online formats [2,3]. In addition, most colleges restricted access to academic buildings and planned to provide, or were already providing, online education to replace current educational materials [4]. In addition, clinical skills sessions and examinations transitioned to a virtual format, resulting in a different learning environment, isolation, and struggles with establishing boundaries between work and home. These changes, in turn, affected the psychological status of the population [2], particularly for undergraduate students. Overall, the COVID-19 pandemic has affected all our lives, especially in the domain of education [5].

Stress and anxiety are relatively common, especially among females and college students [6,7,8]. A number of factors contribute significantly to developing anxiety and depression, such as social and economic status [9]. Students with fewer socio-economic resources appear particularly vulnerable to mental health issues [10,11]. Significantly, stress can impact students’ overall performance and affect their physical and mental health [6]. Previous studies have identified a link between low physical activity and mood disorders, including anxiety and depression [12,13]. Additional reports have linked insomnia to depression in undergraduate students and highlighted the need for insomnia treatment centers to ameliorate mood disorders [14]. The link between poor sleep quality and mood disorders is well established. Insomnia is a common feature in depressed and anxiogenic patients [15].

Previous studies have indicated that during the COVID-19 outbreak, individuals’ physical activity was reduced significantly, indicating that such activity decreases during pandemics [16,17,18,19]. Additionally, physical activity is negatively associated with the risk of developing mood disorders [20,21]. Moreover, maintaining regular physical activity improves the status and quality of sleep [22]. Because our focus was on mental health and sleep quality, we added a measure of physical activity in our study.

To the best of our knowledge, no studies to date have been conducted to examine the multiple factors that may be associated with the negative psychological effects of COVID-19 in Saudi college students during the second wave of the pandemic. Most studies examining the psychological effects of COVID-19 in Saudi college students were either conducted during the first wave of the pandemic or performed on a major-specific domain [23,24,25,26,27,28,29]. Additionally, no study on Saudi undergraduate students has examined the socio-economic link between insomnia, anxiety, and depression. Previous reports have investigated the COVID-19-related mental health and economic burden on Saudi physicians [30], Indian immigrants in Saudi Arabia [31], and the public community [32]. A cross-national longitudinal report found that the number of reported COVID-19 symptoms and students testing positive for COVID-19 was significantly higher in the second wave of the pandemic in most of the six participating countries than in the first wave. Additionally, the number of infected friends or family members was higher in the second wave [33]. In addition, The Saudi Ministry of Health spokesman stated that the second wave of COVID-19 was linked to an increased number of infected personnel following a substantial reduction in COVID-19 positive cases, indicating the second wave represented a significant health burden [34].

Saudi people aged below 25 years, which is the age of most college undergraduate students, make up about 50% of the Saudi population [35], indicating that undergraduate mental health is a significant public health burden. During outbreaks, individuals are put under highly stressful conditions, resulting in a greater risk of developing anxiety. This suggests that students’ mental health may deteriorate over prolonged periods of physical distancing and virtual learning. However, to the best of our knowledge, no study in the Kingdom of Saudi Arabia (KSA) has assessed the link between physical activity, mental health, and insomnia during the COVID-19 pandemic. This research, therefore, explored depression, anxiety, and sleep quality in Saudi undergraduate students. Next, we also examined the effectiveness of physical activities in reducing depression and negative emotions [36,37,38]. Based on the association between these factors, we highlight the need to establish supportive programs. To fulfil our objective, the following specific aims were addressed: (1) Estimating whether the prevalence of physical activity in Saudi students aligns with the WHO recommendation (≥150 min per week). This was achieved using a questionnaire, as described previously [19]. (2) Analyzing the impact of COVID-19 on mental health by examining anxiety and depression in Saudi students using self-reported surveys on generalized anxiety disorder (GAD-7) and a patient health questionnaire (PHQ-9) [39]. (3) Analyzing the impact of COVID-19 on sleep quality using the Pittsburgh Sleep Quality Index (PSQI) [33]. (4) Investigating the possible link between physical activity, anxiety, and depression in light of different variables, such as exposure to COVID-19 [39]. Figure 1 presents the specific aims of the study.

## 2. Materials and Methods

### 2.1. Participants

Participants comprised 207 university students from Saudi Arabia, with a mean age of 23 years (ranging from 18 to 42, M = 22.73, SD = 3.68). The majority were women (89%), single (94%), and had fewer than ten family members (see Table 1). Most participants were from the Central Region (85.51%), and around 40% were interns. Students represented various universities in all geographical regions of Saudi Arabia, different colleges, and all study levels, as listed in Table 1.

### 2.2. Measurement

#### 2.2.1. Exposure to COVID-19

We examined exposure to COVID-19, as described previously [19,33]. The following eight questions were used to assess the consequences of COVID-19: (1) Have you experienced symptoms that could indicate coronavirus infection? (2) Have you been tested for coronavirus? (3) did coronavirus hospitalize you? (4) Did you have to be in strict quarantine for at least 14 days and isolated from loved ones because of the coronavirus infection? (5) Has anyone in your family or friend group been infected with coronavirus? (6) Have any of your relatives died of coronavirus? (7) Have you or a loved one lost their job because of coronavirus? (8) Are you currently experiencing a worsening of your functioning or economic status due to the coronavirus pandemic? Participants answered these questions by choosing *Yes* = 1 or *No* = 0.

#### 2.2.2. Perceived Impact of COVID-19

The perceived negative impact of the COVID-19 (PNIC) was used to assess student’s concerns and perceptions regarding the different situations that might be affected by the COVID-19 pandemic. The PNIC was developed by investigators (but not validated), and has been used in several previous studies [19,33] It addresses five concerns that may emerge during the COVID-19 pandemic: (1) completing the semester and graduating; (2) finding a job and professional development; (3) financial situation (e.g., subsistence during studies); (4) relationships with loved ones and family; (5) relationships with colleagues and friends. Participants responded to each statement on a 5-point Likert scale (ranging from 1 = *Strongly disagree* to 5 = *agree*) [19,33]. Higher PNIC scores indicated more significant COVID-19-related concerns. The PNIC contains two subscales: Socio-Economic Status (SES), which includes three items—completing the semester, finding a job, and financial situation, and Social Relationship (SR), which includes two items—relationships with loved ones and colleagues. The internal reliability for both sub-scales was acceptable, with Cronbach’s α = 0.73 for PNIC-SES and 0.72 for PNIC-SR.

#### 2.2.3. Physical Activity

Physical activity during the COVID-19-related quarantine was assessed as described previously [19], using the following questions: “How many days a week did you exercise physically or pursue sports activities at home or away from home, at the university, in fitness clubs, or at the gym, in the last month?” Responses were reported on an eight-point scale from 0 (*not one day)* to 7 (*seven days a week)*. The second question addressed the average minutes of practice per day (open question). The third question addressed the frequency of activity in the month before the COVID-19 quarantine: “How many days a week did you do physical exercise or sports activities at home or away from home, at the university, in fitness clubs or at the gym, within a month before the general coronavirus quarantine?” The level of physical activity during the previous week was calculated by multiplying the number of days by the number of minutes per day. In the survey, we followed WHO recommendations in categorizing participants into active and inactive groups. Participants who exercised for ≥150 min were included in the group with a sufficient level of physical activity (sufficient), while those exercising for <150 min per week were included in the group with insufficient physical activity for health (insufficient) [40,41].

#### 2.2.4. Anxiety

The Generalized Anxiety Disorder Scale (GAD-7) is a seven-item self-report questionnaire, with high validity, that is used to detect anxiety in the general population [42]. Responses are given on a 4-point Likert scale ranging from 0 (*not at all*) to 3 (*every day*) in the previous two weeks. Total scores of 0 to 21 were used to determine the risk of anxiety [43]. The scale cut-off points are mild (5), moderate (10), and severe anxiety (15) [44]. For logistic regression, the total sample was divided into two groups based on anxiety level, coded as 0 = Low risk of anxiety (GAD-7 scores ≤ 9) and 1 = Moderate risk of anxiety (GAD-7 scores ≥ 10). The Cronbach’s α reliability coefficient for the GAD-7 was 0.98.

#### 2.2.5. Depression

The Patient Health Questionnaire-9 (PHQ-9) is a 9-item self-report measure of depression symptoms that follows the Diagnostic and Statistical Manual of Mental Disorders, Fourth Edition (DSM-IV) criteria [42,45]. Participants were asked to rate the frequency with which they had been troubled by the nine symptoms of depression over the last two weeks on a scale of 0 to 3 (0 = *not at all*, 1 = *several days*, 2 = *more than half the days*, 3 = *nearly every day*) [45]. Total scores were interpreted as indicating depression severity according to the following scale: usual (0–4), mild (5–9), moderate (10–14), moderately severe (15–19), and severe (20–27) [42,44]. Participants in this study were divided into a moderate depression risk group (PHQ-9 scores ≥ 10) and a low depression risk group (PHQ-9 scores ≤ 9) [45]. The Cronbach’s α reliability coefficient for the PHQ-9 was 0.87.

#### 2.2.6. Sleep Quality

The Pittsburgh Sleep Quality Index (PSQI) is a valid questionnaire used to assess sleep habits during the previous month. It consists of 19 self-rated questions and five additional questions rated by a roommate or a bed partner [46,47]. The 19 items are grouped into seven components (rated from 0 to 3) to assess the different variables related to sleep quality. The seven components are: subjective sleep quality, sleep latency, sleep duration, sleep efficiency, sleep disturbance, use of sleep medication, and daytime dysfunction [46]. The total PSQI score ranges from 0 to 21, with higher scores indicating worse sleep quality. The cut-off point of the PSQI was >6 [48], and the Cronbach’s α reliability coefficient was 0.79.

#### 2.2.7. Demographics

The online survey elicited information on the following demographic characteristics: age, gender, marital status, number of family members at home, place of residence in Saudi Arabia (Central, Western, Eastern, Northern, or Southern region), university, college major, and year of study (ranging from the first year to internship).

### 2.3. Study Design

This cross-sectional study was conducted in KSA during the second wave of the outbreak of the COVID-19 pandemic on 6 March and 18 August 2021. The survey was distributed through the most popular social media platforms in Saudi Arabia (Twitter, WhatsApp, Telegram, etc.). Google Forms was used to create a link for the survey, and each participant was invited to take part using a specific link; all participation was voluntary. A consent message indicating that responses were confidential and completely voluntary, along with a clear indication of the project aims, was stated at the beginning of the survey. Consent was obtained by asking participants to confirm their agreement to complete the questionnaire by marking a “yes; I agree to participate” tick box. Ten participants completed the questionnaire, but declined to participate. Ethical approval was granted from the Institutional Review Board at King Saud University in Riyadh, Saudi Arabia (KSU-HE-21-290). A pilot questionnaire was first conducted to validate the survey, following which several amendments were made.

### 2.4. Statistical Analysis

A number of statistical tests were performed to answer the research questions. The McNemar χ^2^ test was conducted to examine changes in physical activity level (sufficient vs. insufficient) between two periods: (1) before and (2) during the COVID-19 pandemic. Associations between anxiety, depression, and sleep quality, exposure to COVID-19, and physical activity were examined using 2 × 2 contingency tables and Pearson’s χ^2^ test of independence, with a ϕ coefficient to assess effect size. Multivariate logistic regression was performed to examine associations between depression (as a dependent variable) and exposure to COVID-19, physical activity, anxiety, and sleep quality (considered as predictor variables), and gender as a potential confounding variable.

Next, we assessed the parametric properties of anxiety, depression, sleep quality, and the perceived negative impact of COVID-19 on daily life, all of which were considered continuous variables. The range of scores, mean (*M*), standard deviation (*SD*), median (*Mdn*), skewness, kurtosis, and Cronbach’s α were calculated for this purpose. A parametric Pearson’s correlation test was then performed to identify associations between variables. Finally, a linear multiple regression analysis was conducted, using the entering method of introducing variables. All statistics were performed using SPSS for Windows, while figures were obtained from JASP for Windows, JASP Team (2020). *JASP* (Version 0.14.1). Computer software. Amstedram, The Netherlands: Department of Psychological Methods, University of Amsterdam.

## 3. Results

### 3.1. Prevalence of Anxiety, Depression, and Exposure to COVID-19

Among the university students, 89 people were at moderate risk of GAD (GAD-7 scores ≥ 10), with a prevalence of 43%. A total of 50% (*n* = 104) of participants also exhibited a moderate depression risk (PHQ-9 scores ≥ 10). The prevalence of poor sleep quality (PSQI > 6) was 66% (*n* = 136). Regarding exposure to COVID-19, 46% (*n* = 95) of students sampled experienced symptoms that could indicate COVID-19 infection (Exposure 1); 61% (*n* = 127) were tested for COVID-19 (Exposure 2); 6% (*n* = 13) were hospitalized by COVID-19 (Exposure 3); 30% (*n* = 61) were in strict quarantine for at least 14 days (i.e., in isolation from loved ones because of the COIVD-19 infection) (Exposure 4); 80% (*n* = 165) stated that some members of their family or friend group were infected with COVID-19 (Exposure 5); 26% (*n* = 53) reported that their relatives had died of COVID-19 (Exposure 6); 16% (*n* = 33) declared that they or a loved one lost their job because of COVID-19 (Exposure 7), and 31% (*n* = 64) experienced worse functioning or lower economic status due to the effects of the pandemic (Exposure 8).

### 3.2. Examining Physical Activity in the Sample of University Students

A contingency table was created to examine and compare changes in physical activity level one month before the pandemic and in the most recent month during the COVID-19 crisis. Overall, only 34% of participants (*n* = 49) were sufficiently involved in physical activity (physical activity ≥ 150 min. per week) during the pandemic, while 21% (*n* = 43) were involved before the pandemic. Only 26 people reported being sufficiently engaged in physical activity both before and during the pandemic. Twenty-three persons exercised insufficiently before the pandemic, but sufficiently during the COVID-19 crisis. Seventeen people were sufficiently involved in physical activity before, but not during the pandemic. The sample of participants with an insufficient level of physical activity both before and during the pandemic consisted of 141 students. The difference between insufficient physical activity level before the pandemic (79.23%) and during (76.33%) was 2.90% (95% CI = −3.18%; 8.98%). These changes were, however, not significant (McNemar χ^2^(1) = 0.90, *p* = 0.34 [corrected for continuity *p* = 0.43]).

### 3.3. The Association between Exposure to the COVID-19 Pandemic and Physical Activity with the Mental Health of University Students

A 2 × 2 contingency table was performed to examine the association between exposure to COVID-19 (no, yes) and physical activity level (sufficient, insufficient), with the risk of anxiety and depression (low risk, moderate risk), and also sleep quality (good, poor). A Pearson’s χ^2^ test of independence was performed to examine significant relationships between dichotomous variables, while ϕ was calculated for effect size. The results are presented in Table 2. Anxiety risk was not related to any of these variables, whereas depression was weakly linked to being in strict quarantine for at least 14 days (χ^2^(1) = 6.49, *p* < 0.05, ϕ = 0.18). Poor sleep quality was related to isolation from loved ones (χ^2^ (1) = 4.94, *p* < 0.05, ϕ = 0.15) and to losing a job because of COVID-19 (χ^2^(1) = 4.52, *p* < 0.05, ϕ = 0.15). Physical activity was not associated with anxiety, depression, or sleep quality.

The logistic regression analysis was performed for all variables, which were considered categorical, including gender (men = 0, women = 1), exposure (no = 0, yes = 1), physical activity (insufficient = 0, sufficient = 1), anxiety (low = 0, moderate = 1), sleep quality (poor = 0, good = 1), and depression (low = 0, moderate = 1). The results (Table 3) indicate that among all variables included in the regression model, only Exposure 4 (being on strict quarantine for at least 14 days in isolation from loved one and family), anxiety, and sleep quality were significant predictors of depression. Exposure to strict quarantine doubled the risk of depression (*OR* = 2.74, *p* < 0.05), moderate or severe anxiety levels increased the risk of depression 15 times (*OR* = 15.48, *p* < 0.001), while sleep quality increased the risk of depression four-fold (*OR* = 4.08, *p* < 0.001). The model of regression can explain 39% of depression variance (Cox–Snell *R*^2^ = 0.392).

### 3.4. Descriptive Statistics

First, the parametric properties of continuous variables (anxiety, depression, sleep quality, and the perceived negative impact of COVID-19 on daily life) were examined, yielding information on the range of scores, mean (*M*), standard deviation (*SD*), median (*Mdn*), skewness, kurtosis, and Cronbach’s α (see Table 4). The value of skewness and kurtosis ranged between +1, indicating that the variables met the assumptions for a parametric statistical analysis.

### 3.5. Associations between Variables

Preliminary correlation analysis was performed to examine associations between variables. The results are presented in Figure 2. A high correlation was found between depression and anxiety, and a moderate correlation between sleep quality and both anxiety and depression, and also between both PNIC scales (SES and SR). SES and SR were weakly related to anxiety and depression, but not sleep quality.

Linear regression was performed to explore predictors of depression among all other variables, including anxiety, sleep quality, exposure to COVID-19 (considered the sum of all eight items), and both PNIC scales SES and SR (Table 5). Anxiety, sleep quality, and PNIC-SES were significant predictors of depression. The model of regression explained 66% of depression variance (*R* = 0.81, *R*^2^ = 0.66, *F*(5, 43) = 16.53, *p* < 0.001).

## 4. Discussion

### 4.1. The Mental Health of Saudi Students

The current study was conducted during the second wave of the pandemic to examine the link between sleep quality, physical activity, and PNIC as predictors of mental health among undergraduate students in Saudi Arabia. It also assessed the need to establish support programs. The results indicated that a significant number, about half of the participants, were at a higher risk of developing depression. A similar risk was reported in those individuals with poor sleep quality. Our multiple linear regression analysis revealed a strong association between depression and anxiety. Furthermore, an association was identified between depression and exposure to COVID-19. About one-third of the participants exhibited a higher risk of depression when undergoing a strict two week quarantine.

Our report revealed that the prevalence of depression is relatively high. However, the difference between insufficient physical activity levels before and during the pandemic was not significant. This aligns with a recent cross-national study, which reported that physical activity was not linked to depression or anxiety in students [49].

The COVID-19 pandemic provoked anxiety and lifestyle changes worldwide. Following the declaration of a Public Health Emergency of International Concern, lockdown and social distancing were implemented in most countries [50]. These preventive measures are linked to psychological well-being [51]. Compounding the psychological impacts most individuals had to endure due to changes in habits and routine, college students had to cope with additional stress factors, such as examinations, academic overload, and adjusting to college life. At college, a sudden and rapid development occurs, both psychologically and physiologically. Developmental psychologists refer to this stage as emerging adulthood [52]. During this phase, challenging stages are encountered, such as long-term life, career, and personal choices, financial independence, personal planning, and becoming a self-reliant adult [53].

Studies published during the COVID-19 pandemic demonstrated that frequent depressive symptoms, anxiety symptoms, suicidal thoughts, and perceived stress are more prevalent among college students than non-students [54]. Another study found that 71% of college students reported increased stress and anxiety due to the COVID-19 pandemic. Only 5% sought help through mental health counseling services [6].

The prevalence of depression reported in our study affected approximately half of the participants, indicating significant potential academic, health, and societal consequences. For example, a recent report using the PHQ-8 scale suggested that the prevalence of depression in Turkish and Polish university students was 60% [55]. Major depressive disorder is a severe disorder, one of the leading disabilities, and has a high comorbidity profile [56,57,58]. A similar magnitude has been reported in Indian medical students [59]. Furthermore, depression increases the risk of suicide by 15% [60], and suicide is the third leading cause of death in young populations [61].

The elevated risk of reported depression could also be related to the gender of the participants. About 90% of our participants were female. It is well-established that females exhibit a higher risk of developing depression and anxiety [62]. The elevated risk may also be due to the nature of the study. Self-reporting studies are confounded by participants’ perceptions and points of view, along with the sample size.

A previous study indicated that the effect of social isolation during confinement on French undergraduate students who lived with their parents was less severe than on those who lived away from their families [63]. Additionally, scores on the self-reported PHQ-9 have indicated that social support from family members, friends, and significant others has a significant impact on the quality of life [64]. Depression due to exposure to a moderate stress levels can be prevented by social support [65]. A United Kingdom social study reported that individuals with high social support scored lower on the PHQ-9, suggesting that social support reduces the risk of developing depression [66]. Additionally, family support promotes coping mechanisms and helps change attitudes toward seeking psychological help [67]. Being away from families and loved ones triggers overwhelming loneliness [68]. Loneliness is linked to passive coping mechanisms [69], and is a unifying feature among psychological diseases, especially mood disorders [70,71].

Our analyses indicated that the prevalence of anxiety affected more than a third of the participants. This pandemic evolved dramatically, and the global community is uncertain about its length and consequences [72]. We found that a substantial number of students exhibited poor sleep quality. Similar findings have been reported in previous studies [73]. Our results also identified a significant association between poor sleep quality and isolation from loved ones and job loss. In line with our findings, prior research has found that poor sleep quality is significantly linked to sociodemographic and interpersonal relationships [74]. A previous study also identified a significant association between insomnia and clinically significant depression and anxiety [75]. Another report linked suicide ideation with sleep distress in university students and athletes [76]. Socio-economic factors are significant stressors during the COVID-19 pandemic. Importantly, family income stability has been found to be a significant stressor in university students [77]. Additionally, a recent cross-national longitudinal study reported that socio-economic deterioration during the COVID-19 pandemic increases the risk of high COVID-19-related post-traumatic stress disorder [33].

Our data indicated that the difference between insufficient physical activity levels before and during the pandemic was not significant, nor was its relationship with mental health status. However, previous studies have reported the opposite [20,21]. This could be attributed to several factors. First, the tools used differ in accuracy and scale fitness [78]; thus, the utilization of other scales might impact the reported results. Different instruments are available for evaluating physical activity, including the International Physical Activity Questionnaire (IPAQ) and the Human Activity Profile (HAP) [79]. A systematic review examining the properties of physical activity questionnaires indicated that existing evidence regarding criterion validity is inconclusive. For example, instruments demonstrated variability in test-retest reliability and validity. Further studies are needed to assess the short-form self-reporting questionnaire [80]. Second, our sample was cross-sectional. The association between physical activity and mental health might require a larger population sample size, as the association was reported to be small and inconsistent [81].

A confounding factor in our research was that a significant number of participants were female, preventing us from examining gender differences. A similar observation was reported in other studies [26,82]. A recent systematic review indicated that the COVID-19 pandemic negatively affects the health-related quality of life in children and/or adolescents. Notably, the review found that only a couple of studies examined gender differences, suggesting a lack of gender-based studies in this field. The first report indicated that gender differences existed before the pandemic in young German populations. However, these differences were not significant during the pandemic. Similarly, during the pandemic, no gender differences existed in the health-related quality of life of young Spanish populations [83].

### 4.2. Institutional Mental Health Support Systems

College and university counseling services have been established in different areas around the world in an effort to help students overcome the various problems they may face while at college. For example, in the United States, a typical university counseling center staff member holds a license to practice, usually as a psychologist, and anxiety and depression are the most common concerns presented by students [84]. The University of Washington Counseling Center is one example of a counseling service that provides a variety of options for students seeking assistance with stress and mental health issues. Their service offers workshops, individual counseling, group counseling, and referrals for students interested in long-term or specialized counseling. They also provide multiple self-help online materials on their web pages, such as apps, articles, and podcasts that provide information on different topics, but these are not intended to be a substitute for counseling [85]. Similar services are also offered by the counseling and mental health support at Northumbria University in Newcastle [86]. While counseling services in universities across the United States and the United Kingdom seem to offer different styles of counseling services, almost all provide a hotline service for emergencies and immediate assistance for situations such as suicidal ideation, extreme psychological distress, or being assaulted. 

Regarding local counseling services, the Kingdom of Saudi Arabia has established the National Center for Mental Health Promotion, which aims to promote mental health awareness in the community, help people obtain treatment and rehabilitation services, and promote access to a better life. Furthermore, to address the growing number of mental health issues among college students, the National Center for Mental Health Promotion has issued a request to universities in all regions of the Kingdom to establish groups to promote mental health. In response, many universities in the area have established student counseling services, some of which offer appointments for personal counseling, such as the Effat University Student Counseling Services, Imam Abdulrahman bin Faisal University Counseling Center, Imam Muhammad bin Saud Islamic University Psychological and Social Counseling Unit, Princess Nourah bint Abdulrahman University Family Consulting Center, and King Saud University Psychological Services Unit (as part of the department of psychology at the college of education). In addition, universities such as Saudi Electronic University and King Saud University have established committees to organize seminars and awareness programs. 

Most Saudi universities offer counseling services and provide all the contact information needed to book an appointment. However, some essential aids are missing, such as adequate online resources and self-help and crisis hotline materials. Counseling services are vital for college students who exhibit higher levels of distress and need mental health counseling. It is crucial to efficiently execute the counseling services for college students to reduce attrition, improve academic and psychosocial performance, and reduce the risk of onset of a wide range of chronic physical conditions resulting from mental illnesses.

### 4.3. The Utilities and Arrangements for COVID-19 Testing in Saudi Arabia

Our results highlighted that numerous populations had been tested for COVID-19 (Exposure 2). Saudi Arabia has established guidelines and implemented proactive preventive measures for the preparedness, detection, testing, tracing, isolation, and treatment of COVID-19 [87], making it one of the first countries to implement early preventive measures to control the spread of the virus [88]. In addition to Saudi Arabia’s Ministry of Health, military hospitals and other government-sponsored institutions provide free healthcare services for the general population. COVID-19 testing and treatment have also been offered for all citizens, legal, and illegal residents, free of charge and without any consequences [87]. Surveillance and mass testing are crucial measures initiated by Saudi Arabia in the early phase of the pandemic. According to a recent study, testing is a critical factor in slowing the spread of the disease. Countries with a high rate of positive cases are less likely to have tested widely enough to detect positive COVID-19 cases at an early stage [89]. These data imply a clear link between a country’s choice to conduct mass screening tests and the total number of deaths and infected cases.

One study demonstrated that the best course of action for community health is to begin testing as soon as possible, which will allow for early discovery of infection, treatment, and isolation of infected individuals to prevent disease transmission [89]. In Saudi Arabia, mass screening was implemented in stages, with the first stage focusing on conducting field testing in heavily populated areas [90]; this resulted in the identification of a large number of confirmed cases and their contacts, and the implementation of necessary preventive measures to avoid future spread [87].

The Saudi government has established and implemented both “Taakkad” centers and Tetamman clinics [87,90]. High laboratory capacity and preparedness were necessary to manage and analyze COVID-19 samples. To optimize the national initiatives targeted at preventing the spread of COVID-19, the Ministry of Health collaborated with the Saudi Center for Disease Prevention and Control (SCDC), regional laboratories, and various reference laboratories in the health sector. In addition to laboratories with a capacity estimated at over 80,000 tests per day, Saudi Arabia also signed a contract with China to provide nine million COVID-19 diagnostic tests, along with devices, specialists, and specialized technicians, and the establishment of six large regional laboratories distributed across various regions of the Kingdom, including a mobile laboratory with the capacity to perform 10,000 tests per day [87].

### 4.4. Study Limitations

The first limitation of the current study is that most of the participants were from the Central Region. Moreover, most of the sample subjects were single females. Significantly, some studies have indicated that being single and female are risk factors for anxiety and stress. The output of this study is significant, and the sample size is adequate, but the number of participants is not considered large. However, it is important to note that a similar sample size was acceptable in both GAD-7 [91,92,93] and PHQ-9 studies [94,95]. Our study also reported results that are consistent with published studies. The second limitation of this study relates to the measurement method, namely the self-report online questionnaires that relied on convenience sampling. Nevertheless, this was the only way in which mental health could be investigated during the lockdown.

## 5. Conclusions

Our study highlighted the significant prevalence of anxiety, poor sleep quality, and depression among undergraduate students. It further examined the impact of isolation from loved ones and job loss. This report underlines the socio-economic impact of this pandemic and the overwhelming prevalence of depression. Based on the findings, the researchers recommend establishing and developing institutional mental health preventive and support programs focused on improving sleep quality and decreasing anxiety and depression among university students.

## Figures and Tables

**Figure 1 ijerph-19-06262-f001:**
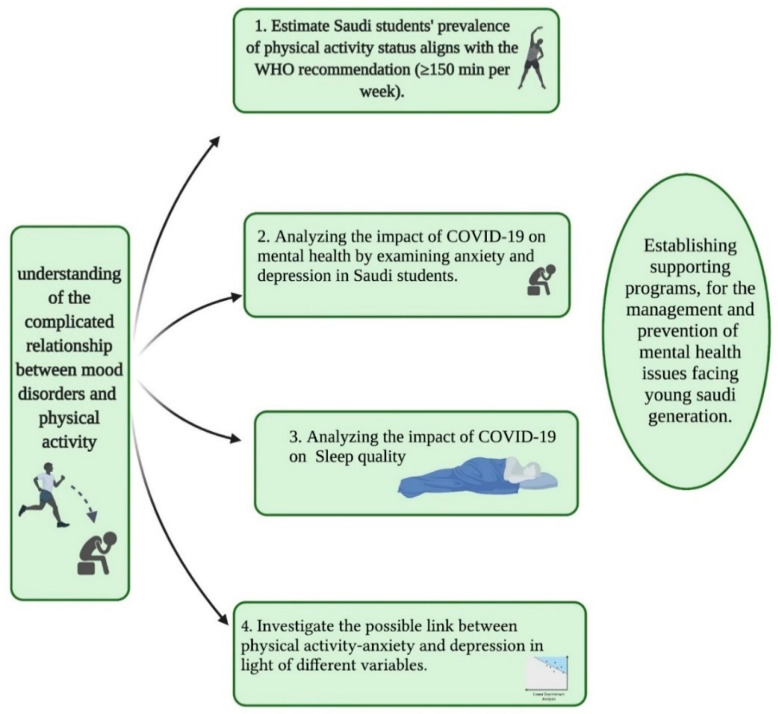
The specific aims of the study.

**Figure 2 ijerph-19-06262-f002:**
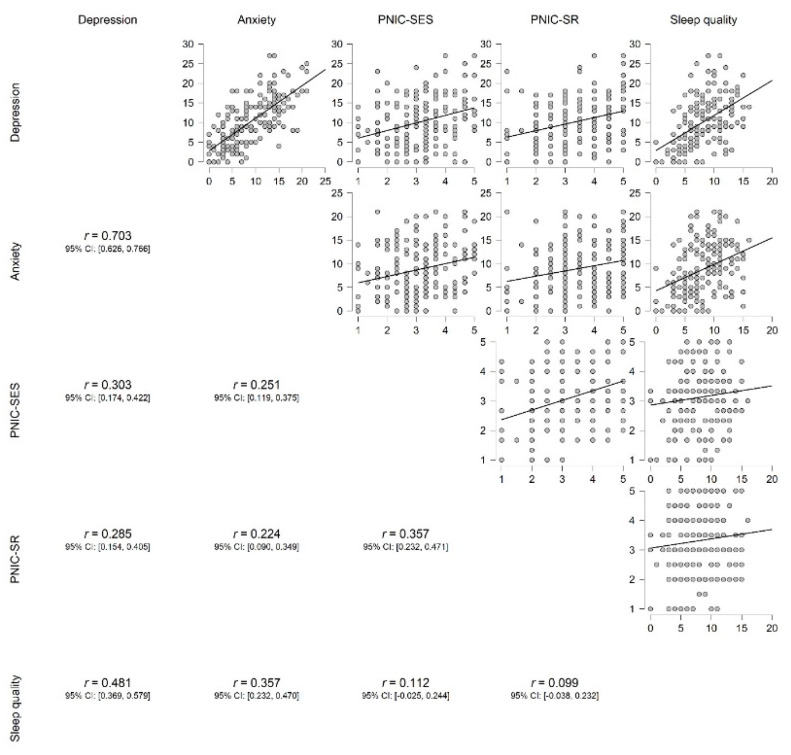
Scatter plots for associations between study variables. PNIC-SES = Socio-Economic Status of the Perceived Negative Impact of COVID-19; PNIC-SR = Social Relationships of the Perceived Negative Impact of COVID-19.

**Table 1 ijerph-19-06262-t001:** Demographic characteristics of the sample.

Variable	Categories	*n*	%
Gender	Women	185	89.37
	Men	22	10.63
Marital status	Single	194	93.72
	Married	13	6.28
Number of family members	1	6	2.90
	2	5	2.42
	3	10	4.83
	4	15	7.25
	5	28	13.53
	6	34	16.43
	7	46	22.22
	8	32	15.46
	9	15	7.25
	10	6	2.90
	11	6	2.90
	12	3	1.45
	13	1	0.48
Geographical region	The Central Region (Riaydh, Qasim)	177	85.51
	The Western Region (Mecca, Medina, Jeddah)	10	4.83
	The Eastern Region (Damam, Khafji, Alhasa)	8	3.86
	The Southern Region (Asir, Najran, Jizan)	3	1.45
	The Northern Region (Tabuk, Jouf, Hail)	9	4.35
Study year	Internship	82	39.61
	First	9	4.35
	Second	14	6.76
	Third	27	13.04
	Fourth	39	18.84
	Fifth	25	12.08
	Sixth	11	5.31

**Table 2 ijerph-19-06262-t002:** The associations of anxiety and depression with exposure to COVID-19 and physical activity *N* = 207).

Variable	Anxiety						Depression						Sleep Quality					
	Low Risk		Moderate Risk				Low Risk		Moderate Risk				Good		Poor			
	*n*	%	*n*	%	χ^2^(1)	ϕ	*n*	%	*n*	%	χ^2^(1)	ϕ	*n*	%	*n*	%	χ^2^(1)	ϕ
Exposure 1					1.86	−0.09					0.58	−0.05					0.22	0.03
No	59	52.68	53	47.32			53	47.32	59	52.68			40	56.34	72	52.94		
Yes	59	62.11	36	37.89			50	52.63	45	47.37			31	43.66	64	47.06		
Exposure 2					0.21	−0.03					0.83	0.06					1.15	0.28
No	44	55.00	36	45.00			43	53.75	37	46.25			31	43.66	49	36.03		
Yes	74	58.27	53	41.73			60	47.24	67	52.76			40	56.34	87	63.97		
Exposure 3					0.06	0.02					2.00 _a_	0.10					2.20 _a_	0.10
No	111	57.22	83	42.78			99	51.03	95	48.97			69	97.18	125	91.91		
Yes	7	53.85	6	46.15			4	30.77	9	69.23			2	2.82	11	8.09		
Exposure 4					0.73	0.06					6.49 _a_*	0.18					4.94 _a_*	0.15
No	86	58.9	60	41.10			81	55.48	65	44.52			57	80.28	89	65.44		
Yes	32	52.46	29	47.54			22	36.07	39	63.93			14	19.72	47	34.56		
Exposure 5					1.05 _a_	−0.07					1.82 _a_	−0.09					0.02 _a_	−0.01
No	21	50.00	21	50.00			17	40.48	25	59.52			14	19.72	28	20.59		
Yes	97	58.79	68	41.21			86	52.12	79	47.88			57	80.28	108	79.41		
Exposure 6					0.80 _a_	−0.06					0.01	0.01					0.37	−0.04
No	85	55.19	69	44.81			77	50.00	77	50.00			51	71.83	103	75.74		
Yes	33	62.26	20	37.74			26	49.06	27	50.94			20	28.17	33	24.26		
Exposure 7					0.10 _a_	0.02					4.24 _a_	0.14					4.52 _a_*	0.15
No	100	57.47	74	42.53			92	52.87	82	47.13			65	91.55	109	80.15		
Yes	18	54.55	15	45.45			11	33.33	22	66.67			6	8.45	27	19.85		
Exposure 8					2.77	0.11					1.34	0.08					1.57 _a_	0.09
No	87	60.84	56	39.16			75	52.45	68	47.55			53	74.65	90	66.18		
Yes	31	48.44	33	51.56			28	43.75	36	56.25			18	25.35	46	33.82		
PA					0.00	0.00					0.28	0.04					3.20	0.12
Sufficient	28	57.14	21	42.86			26	53.06	23	46.94			22	30.99	27	19.85		
Insufficient	90	56.96	68	43.04			77	48.73	81	51.27			49	69.01	109	80.15		

*Note*: Exposure = Exposure to COVID-19 to assess the consequences of COVID-19. Exposure 1 = symptoms of COVID-19 infection; Exposure 2 = test for COVID-19; Exposure 3 = hospitalization; Exposure 4 = strict quarantine for at least 14 days; Exposure 5 = family or friend infected; Exposure 6 = death of a loved one or relative; Exposure 7 = job loss; Exposure 8 = a worsening economic status; PA = Physical activity during the coronavirus-related quarantine; _a_ = the Fisher exact test statistical value, used due to the small sample size; * *p* < 0.05.

**Table 3 ijerph-19-06262-t003:** Results of logistic regression analysis for depression.

Variable	*b*	*SE b*	β	*AOR*	95% *CI*		Wald Test		
					*LL*	*UL*	*z*	χ^2^(1)	*p*
Intercept	−1.43	0.80	0.03	0.24	0.05	1.15	−1.79	3.21	0.073
Gender	−1.09	0.66	−0.34	0.34	0.09	1.21	−1.67	2.77	0.096
Exposure 1	−0.58	0.44	−0.29	0.56	0.24	1.33	−1.32	1.73	0.188
Exposure 2	0.40	0.43	0.20	1.49	0.64	3.48	0.93	0.86	0.355
Exposure 3	0.62	0.88	0.15	1.86	0.33	10.32	0.71	0.50	0.481
Exposure 4	1.01	0.47	0.46	2.74	1.08	6.94	2.12	4.51	0.034
Exposure 5	−0.50	0.49	−0.20	0.61	0.23	1.57	−1.03	1.07	0.302
Exposure 6	−0.08	0.47	−0.04	0.92	0.36	2.32	−0.18	0.03	0.858
Exposure 7	0.87	0.56	0.32	2.40	0.80	7.17	1.56	2.44	0.118
Exposure 8	−0.41	0.43	−0.19	0.66	0.29	1.54	−0.96	0.92	0.339
PA	0.56	0.50	0.24	1.74	0.66	4.64	1.11	1.24	0.265
Anxiety	2.74	0.41	1.36	15.48	6.98	34.32	6.74	45.44	<0.001
Sleep quality	1.41	0.42	0.67	4.08	1.81	9.20	3.39	11.46	<0.001

*Note*. Exposure = Exposure to COVID-19 to assess the consequences of COVID-19. Exposure 1 = symptoms of COVID-19 infection; Exposure 2 = test for COVID-19; Exposure 3 = hospitalization; Exposure 4 = strict quarantine for at least 14 days; Exposure 5 = family or friend infected; Exposure 6 = death of a loved one or relative; Exposure 7 = job loss; Exposure 8 = a worsening economic status; PA = physical activity during the COVID-19-related quarantine. *SE* = standard deviation, *AOR* = adjusted odds ratio, *CI* = confidence interval, *LL* = lower level, *UL* = upper level.

**Table 4 ijerph-19-06262-t004:** Descriptive statistics for study variables.

Scale	Range	*M*	*SD*	*Mdn*	Skewness	Kurtosis	Cronbach’s α
PHQ-9	0–27	10.14	6.45	10	0.30	−0.54	0.87
GAD-7	0–21	8.83	5.48	8	0.24	−0.88	0.89
PSQI	0–16	8.12	3.48	8	0.05	−0.66	0.79
PNIC-SES	1–5	3.12	1.02	3	−0.13	−0.52	0.73
PNIC-SR	1–5	3.32	1.11	3	−0.21	−0.69	0.72

*Note*. GAD-7 = a seven-item scale of General Anxiety Disorder; PHQ-9 = a nine-item scale of Patient Health; PSQI = the Pittsburgh Sleep Quality Index; PNIC-SES = Socio-Economic Status of the Perceived Negative Impact of COVID-19; PNIC-SR = Social Relationships of the Perceived Negative Impact of COVID-19.

**Table 5 ijerph-19-06262-t005:** Results of linear regression analysis for depression (*N* = 207).

Variable	*b*	*SE b*	β	*t*	*p*	95% CI	
						*LL*	*UL*
Constant	−6.12	2.71		−2.26	0.029	−11.59	−0.65
Anxiety	0.42	0.16	0.33	2.73	0.009	0.11	0.74
Sleep quality	0.72	0.27	0.32	2.73	0.009	0.19	1.26
Exposure to COVID-19	−0.03	0.39	−0.01	−0.07	0.942	−0.82	0.76
PNIC-SES	1.72	0.72	0.26	2.39	0.021	0.27	3.17
PNIC-SR	0.81	0.85	0.12	0.96	0.344	−0.90	2.53

*Note*. PNIC-SES = Socio-Economic Status of the Perceived Negative Impact of COVID-19; PNIC-SR = Social Relationships of the Perceived Negative Impact of COVID-19.

## Data Availability

Data are available upon reasonable request.

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
