# Peer review of "Examining Anxiety, Sleep Quality, and Physical Activity as Predictors of Depression among University Students from Saudi Arabia during the Second Wave of the COVID-19 Pandemic"

_ijerph, 2022, doi:10.3390/ijerph19106262_

Round 1
Reviewer 1 Report
So far into the COVID-19 pandemic, much data has already been collected about the relationship between various indicators of psychological distress and COVID-19 exposure. Some of the nuances may still be explored, and there is a case for focusing on specific populations. The population of undergraduate students would be one example. However, any such study requires both substantially large samples as well as diverse samples that provide the reader with some confidence about representativeness. Using a small convenience sample does not provide any convincing insights. Additionally, there are no new contributions in terms of our understanding of how psychological variables are related. No particular model was tested, and the conclusions rely on speculation based on the results from correlating a few variables in a cross-sectional design.
Author Response
So far into the COVID-19 pandemic, much data has already been collected about the relationship between various indicators of psychological distress and COVID-19 exposure. Some of the nuances may still be explored, and there is a case for focusing on specific populations. The population of undergraduate students would be one example. However, any such study requires both substantially large samples as well as diverse samples that provide the reader with some confidence about representativeness. Using a small convenience sample does not provide any convincing insights. Additionally, there are no new contributions in terms of our understanding of how psychological variables are related. No particular model was tested, and the conclusions rely on speculation based on the results from correlating a few variables in a cross-sectional design.
Thank you. In terms of sample size and diversity, it is a good point, and we already mentioned it in the study limitation. In fact, we tried to gather more samples during the study-specific time (second wave), but we couldn't, and we found other studies published with similar sample sizes; thus, we proceeded with the work.
Study limitations
The first limitation of the current study is that most of the participants were from the central region. Also, most of the sample were single females. Significantly, some studies have indicated that being single and female are risk factors for anxiety and stress. The output of this study is significant, and the sample size is adequate, but the number of participants is not considered large. However, it is important to note that a similar sample size was acceptable in both GAD-7 (15-17) and PHQ-9 studies (18, 19). Also, our reported results are consistent with published studies. The second limitation of this study relates to the measurement method, namely the self-report online questionnaires that relied on convenience sampling. Nevertheless, this was the only way in which mental health could be investigated during the lockdown.
Regarding the study contributions, we added this paragraph highlighting the study's novelty to the introduction, focusing on the Saudi community. (L52-66)
To the best of our knowledge, no studies to date have be conducted to examine the multiple factors that may be associated with the negative psychological effects of COVID-19 in Saudi college students during the second wave of the pandemic. Most studies examining the psychological effects of COVID-19 in Saudi college students were either conducted during the first wave of the pandemic or performed on a major-specific domain (1-7). Additionally, no study on Saudi undergraduate students has examined the socio-economic link between insomnia, anxiety, and depression. Previous reports have investigated the COVID-19 related mental health and economic burden on Saudi physicians (8), Indian immigrants in Saudi Arabia (9), and the public community (10). A cross-national longitudinal report showed that the number of reported COVID-19 symptoms and the number of COVID-19 tested students are significantly higher in the second wave of the COVID-19 pandemic in most of the six participating countries compared to the first wave. Additionally, the number of infected friends or family was higher in the second wave (11). Besides, The Saudi Ministry of Health Spokesman stated that the second wave of COVID-19 is linked to an increased number of infected personnel following a substantial reduction in COVID-19 positive cases, indicating the second wave represented a significant health burden(12).
Reviewer 2 Report
I think this manuscript will be important findings to overwhelm psychological distress or mental health deterioration among university students during the COVID-19 pandemic. I would like you to point out several issues in this manuscript that should be revised.
L67: I recommend you be clear about your research hypothesis in this study. (e.g. in our hypothesis, high physical activities could prevent mental health issues, therefore, we examined the link between them and highlight the need to establish supportive programs.)
Figure 1: It needs a detailed explanation of Figure 1. Especially, why the authors focused on physical activities to prevent psychological distress or mental health deterioration. Could you please mention these backdrops while using references?
L75: Could you show the reason why male students were extremely low in number to participate in this survey? Regarding this study's representatively, it might be a critical issue.
L99: Is the perceived negative impact of the COVID-19 (PNIC) a validated scale or investigators' developed query? Please show it in this method section.
Table2: The right side of the table isn't shown because of cutting off.
L294: The Covid-19 ---> The COVID-19
L351: "Institutional mental health support system" Is the description in this paragraph based on the findings of this survey? If not, please clarify why you need to mention it.
Discussion & Conclusion: In the beginning, physical activities (PA) will play an important role preventing psychological distress or deteriorating mental health status as shown the Figure 1. However, there is little mention about PA in the discussion and conclusion section. Could you please show the reason why physical activity did not have a significant relationship with mental mealth status under the COVID-19 pandemic.
Author Response
I think this manuscript will be important findings to overwhelm psychological distress or mental health deterioration among university students during the COVID-19 pandemic. I would like you to point out several issues in this manuscript that should be revised.
L67: I recommend you be clear about your research hypothesis in this study. (e.g. in our hypothesis, high physical activities could prevent mental health issues, therefore, we examined the link between them and highlight the need to establish supportive programs.)
Thank you. A sentence has been added (L82-84).
Secondly, we also examined the effectiveness of physical activities in reducing depression and negative emotions (20-22).
Figure 1: It needs a detailed explanation of Figure 1. Especially, why the authors focused on physical activities to prevent psychological distress or mental health deterioration. Could you please mention these backdrops while using references?
Thank you. We added a rationale for including physical activity to the study (L50-56).
Previous studies have indicated that during the COVID-19 outbreak, individuals' physical activity reduced significantly, indicating that such activity decreases during pandemics (23-26). Additionally, physical activity is negatively associated with the risk of developing mood disorders (27, 28). Moreover, maintaining regular physical activity improves the status and quality of sleep (29). Because our focus was on mental health and sleep quality, we added a measure of physical activity in our study.
Also, we added more clarification to the text (L70-90).
To fulfil our objective, the following specific aims were addressed: 1) Estimating whether the prevalence of physical activity in Saudi students' aligns with the WHO recommendation (≥150 min per week). This was achieved using a questionnaire, as described previously (30). 2) Analyzing the impact of COVID-19 on mental health by examining anxiety and depression in Saudi students using self-reported surveys on generalized anxiety disorder (GAD-7) and a patient health questionnaire (PHQ-9) (30). 3) Analyzing the impact of COVID-19 on sleep quality using the Pittsburgh Sleep Quality Index (PSQI) (31). 4) Investigating the possible link between physical activity, anxiety, and depression in light of different variables, such as exposure to COVID-19 (30). Figure 1 presents the specific aims of the study.
L75: Could you show the reason why male students were extremely low in number to participate in this survey? Regarding this study's representatively, it might be a critical issue.
We don’t have a specific explanation, but we noticed that the drive to fill and participate in a study is less in male students. We included this point in the discussion. (L407-416)
A confounding factor in our research was that a significant number of participants were female, preventing us from examining gender differences. A similar observation was reported in other studies (4, 13). A recent systematic review indicated that the COVID-19 pandemic negatively affects the health-related quality of life in children and/or adolescents. Notably, the review found that only a couple of studies examined gender differences, suggesting a lack of gender-based studies in this field. The first report indicated that gender differences existed before the pandemic in young German populations. However, these differences were not significant during the pandemic. Similarly, during the pandemic, no gender differences existed in the health-related quality of life of young Spanish populations (14).
L99: Is the perceived negative impact of the COVID-19 (PNIC) a validated scale or investigators' developed query? Please show it in this method section.
- Thank you for the question. We added to the Method section the following explanation; “The PNIC was developed by investigators (but not validated), and used in some previous studies (26, 32).”
Table2: The right side of the table isn't shown because of cutting off.
- Thank you for the comment. We corrected the mistake.
L294: The Covid-19 ---> The COVID-19
Done.
L351: "Institutional mental health support system" Is the description in this paragraph based on the findings of this survey? If not, please clarify why you need to mention it.
The primary aim of this study is to highlight the need to establish supporting programs. Thus we believe that reviewing the current status of supporting programs would benefit our research purposes.
Discussion & Conclusion: In the beginning, physical activities (PA) will play an important role preventing psychological distress or deteriorating mental health status as shown the Figure 1. However, there is little mention about PA in the discussion and conclusion section. Could you please show the reason why physical activity did not have a significant relationship with mental health status under the COVID-19 pandemic.
Addressed, discussion (L394-406)
Our data indicated that the difference between insufficient physical activity levels before and during the pandemic was not significant, nor was its relationship with mental health status. However, previous studies have reported the opposite (27, 28). This could be attributed to several factors. First, the tools used differ in accuracy and scale fitness (33), thus the utilization of other scales might impact the reported results. Different instruments are available for evaluating physical activity, including the International Physical Activity Questionnaire (IPAQ) and the Human Activity Profile (HAP)(34). A systematic review examining the properties of physical activity questionnaires indicated that existing evidence regarding criterion validity is inconclusive. For example, instruments demonstrated variability in test-retest reliability and validity. Further studies are needed to assess the short-form self-reporting questionnaire (35). Second, our sample was cross-sectional. The association between physical activity and mental health might require a larger population sample size, as the association was reported to be small and inconsistent (36).
Reviewer 3 Report
Dear authors, thanks for allowing me to review this interesting work. I have some queries/suggestions :
1) At line 42, the authors stated that students from low socio-economic background had poor mental health. The following reference also discusses the relationship between socio-economic status and coping strategies and should be included as a reference :
Francis B, Soon Ken C, Yit Han N, et al. Religious coping during the COVID-19 pandemic: gender, occupational and socio-economic perspectives among Malaysian frontline healthcare workers. Alpha Psychiatry. 2021;22(4):194-199.
2) Line 40, the authors mentioned that university students are often stressed and anxious. The authors are guided to the following references which report global findings on depression and anxiety among university students :
Francis, B., Gill, J. S., Yit Han, N., Petrus, C. F., Azhar, F. L., Ahmad Sabki, Z., ... & Sulaiman, A. H. (2019). Religious coping, religiosity, depression and anxiety among medical students in a multi-religious setting. International journal of environmental research and public health, 16(2), 259.
Lew, B., Huen, J., Yu, P., Yuan, L., Wang, D. F., Ping, F., ... & Jia, C. X. (2019). Associations between depression, anxiety, stress, hopelessness, subjective well-being, coping styles and suicide in Chinese university students. PloS one, 14(7), e0217372.
3) Line 64 and 65 seem incoherent. Please correct.
4) The introduction should include a brief literature review on the impact of physical activity and insomnia on the mental health of university students.
5) Table 2 is unclear : what do the exposures relate to and why are there 8 exposures ?
6) Instead of depicting the relationship of the variables using a scatter plot, the authors should present them more clearly in a multivariate logistic regression table. Please revise.
7) There are several grammatical errors throughout the manuscript. The authors are encouraged to proofread the manuscript before resubmitting.
Author Response
Dear authors, thanks for allowing me to review this interesting work. I have some queries/suggestions :
1) At line 42, the authors stated that students from low socio-economic background had poor mental health. The following reference also discusses the relationship between socio-economic status and coping strategies and should be included as a reference :
Francis B, Soon Ken C, Yit Han N, et al. Religious coping during the COVID-19 pandemic: gender, occupational and socio-economic perspectives among Malaysian frontline healthcare workers. Alpha Psychiatry. 2021;22(4):194-199.
Addressed.
2) Line 40, the authors mentioned that university students are often stressed and anxious. The authors are guided to the following references which report global findings on depression and anxiety among university students :
Francis, B., Gill, J. S., Yit Han, N., Petrus, C. F., Azhar, F. L., Ahmad Sabki, Z., ... & Sulaiman, A. H. (2019). Religious coping, religiosity, depression and anxiety among medical students in a multi-religious setting. International journal of environmental research and public health, 16(2), 259.
Lew, B., Huen, J., Yu, P., Yuan, L., Wang, D. F., Ping, F., ... & Jia, C. X. (2019). Associations between depression, anxiety, stress, hopelessness, subjective well-being, coping styles and suicide in Chinese university students. PloS one, 14(7), e0217372.
Addressed.
3) Line 64 and 65 seem incoherent. Please correct.
The introduction has been re-organized into:
A cross-national longitudinal report found that the number of reported COVID-19 symptoms and students testing positive for COVID-19 was significantly higher in the second wave of the pandemic in most of the six participating countries than in the first wave. Additionally, the number of infected friends or family was higher in the second wave (11). In addition, The Saudi Ministry of Health Spokesman stated that the second wave of COVID-19 was linked to an increased number of infected personnel following a substantial reduction in COVID-19 positive cases, indicating the second wave represented a significant health burden (12). Saudi people aged below 25 years, which is the age of most college undergraduate students, make up about 50% of the Saudi population (37), indicating that undergraduate mental health is a significant public health burden.
4) The introduction should include a brief literature review on the impact of physical activity and insomnia on the mental health of university students.
Thank you for your point, we added a rationale for including physical activity to the study (L50-56).
Previous studies have indicated that during the COVID-19 outbreak, individuals' physical activity reduced significantly, indicating that such activity decreases during pandemics (23-26). Additionally, physical activity is negatively associated with the risk of developing mood disorders (27, 28). Moreover, maintaining regular physical activity improves the status and quality of sleep (29). Because our focus was on mental health and sleep quality, we added a measure of physical activity in our study.
5) Table 2 is unclear : what do the exposures relate to and why are there 8 exposures ?
- Thank you for the question. We described the exposure to COVID-19 in the method section in more details, but indeed, the table is not very informative. We added the explanation about the specific exposure in each item to the footer of Table 2 and Table 3: Exposure = Exposure to COVID-19 to assess the consequences of COVID-19: “Exposure 1 = symptoms of coronavirus infection, Exposure 2 = test for coronavirus, Exposure 3 = hospitalizing, Exposure 4 = strict quarantine for at least 14 days, Exposure 5 = family or friend infected, Exposure 6 = death of a loved one or relatives, Exposure 7 = job loss, Exposure 8 = a worsening economic status”
6) Instead of depicting the relationship of the variables using a scatter plot, the authors should present them more clearly in a multivariate logistic regression table. Please revise.
- Thank you for the comment, we added the logistic regression table to the Results section (see Table 3), as suggested.
- However, the scatter plot was presented as a preliminary analysis for linear regression, for selected variables considered as a continuous instead of categorical (i.e., anxiety, depression, sleep quality, PNIC), so the Pearson’s r correlations and multivariate linear regression model were presented. The scatter plot demonstrate correlations between variables, including such statistics as Pearson’s r, 95% confidence interval (CI), density of scores distribution and the slop of regression line. All statistics clearly and informatively shows associations between variables, therefore we prefer to not delete this analysis.
7) There are several grammatical errors throughout the manuscript. The authors are encouraged to proofread the manuscript before resubmitting.
The manuscript has been revised extensively by an English editor.
Round 2
Reviewer 1 Report
Thank you
Reviewer 3 Report
Thanks for the revisions. I have no further queries.